# Success Factors for Water Safety Plan Implementation in Small Drinking Water Supplies in Low- and Middle-Income Countries

**Jo Herschan** [1,*]**, Bettina Rickert** [2]**, Theresa Mkandawire** [3]**, Kenan Okurut** [4]**, Richard King** [1] **, Susan J. Hughes** [1]**, Dan J. Lapworth** [5] **and Katherine Pond** [1]

1   Centre for Environmental Health and Engineering (CEHE), Department of Civil and Environmental Engineering, University of Surrey, Guildford, Surrey GU2 5XH, UK; r.a.king@surrey.ac.uk (R.K.); sue.hughes@surrey.ac.uk (S.J.H.); k.pond@surrey.ac.uk (K.P.)
2   German Environment Agency (UBA), Schichauweg 58, 12307 Berlin, Germany; Bettina.Rickert@uba.de
3   Department of Civil Engineering, University of Malawi—The Polytechnic, Private Bag 303, Blantyre, Malawi; tmkandawire@poly.ac.mw
4   Department of Civil and Building Engineering, University of Kyambogo, Kyambogo Road, Kiwatule-Banda, Kampala, Uganda; ken_okurut@yahoo.com
5   British Geological Survey, Wallingford OX10 8BB, UK; djla@bgs.ac.uk
*   Correspondence: j.herschan@surrey.ac.uk

**Abstract:** Water Safety Plan (WSP) implementation has the potential to greatly improve, commonly very challenging and resource limited, small drinking water supplies. Although slower than in urban or high-income settings, the uptake of WSPs in low- and middle-income countries (LMICs) is accelerating. Understanding the factors which will make a WSP successful will further improve efficient uptake and assist with its long-term sustainability. Based on an extensive literature search using the Preferred Reporting Items for Systematic Reviews and Meta-Analyses (PRISM-A) methodology, 48 publications, including case studies and guidance documentation, formed the basis of this review. These were analysed using inductive and deductive coding methods to (i) identify the success factors applicable to WSP implementation in small drinking water supplies in LMICs and (ii) to investigate which factors are more or less critical depending on the geography and level of development of the implementing country. Key challenges identified during the review process were also noted. A comparison of these success factors was made with those identified from high-income and urban settings. The three most important success factors identified are the development of technical capacity, community engagement, and monitoring and verification. Factors specific to small drinking water supplies in LMICs include support from non-government organisations, integration into existing water sanitation and hygiene (WASH) programs, simplicity, and community engagement. Certain factors, such as adaptability, the use of guidance documentation, international collaboration, the role of pilot studies, knowledge sharing, and stakeholder involvement are applicable to all WSP settings. Due to the specific challenges faced by small drinking water systems and the limited number of original research publications on this topic, this study highlights the need for further data collection and research focused on success factors in these settings. It is anticipated that the consideration of the success factors identified in this study will assist implementers in improving the uptake and long-term sustainability of WSPs in small drinking water supplies in low- and middle-income settings.

**Keywords:** drinking water; enabling environment; small drinking water supplies; success factor; sustainable development goals; water safety plan; water sanitation and hygiene (WASH)

## 1. Introduction

The Water Safety Plan (WSP) approach to managing drinking water supplies, through systematic and proactive risk assessment and risk management, has been gathering pace since 2004, when the World Health Organization (WHO) provided guidance for their development and implementation in the Guidelines for Drinking Water Quality [1]. The WSP approach required a change in mindset away from the reliance on end-product testing of drinking water quality to the combined use of drinking water quality testing and risk management. Although still necessary, solely relying on end-product testing of drinking water quality provides limited information, as results are retrospective based on just a snapshot in time, by which point the water is likely to have already been consumed [2]. Additionally, drinking water quality testing requires access to physical and financial resources, so reducing reliance on testing is especially pertinent in low-resource settings [3].

Conforming to the general principles of risk assessment and risk management, the WSP methodology is flexible, and approaches can and should be tailored to suit the individual context [4,5]. This flexibility is reflected in the approach being used in various settings with diverse challenging conditions, from regions of conflict [6,7] and refugee camps [8] to cruise ships [9]. Additionally, WSP versatility is demonstrated by the wide reach of application of the approach, with WSPs now implemented in every region globally [10].

Initially, uptake of WSPs was greatest in urban settings and by water utilities in developed countries [11,12]. Concerns were raised that small drinking water supplies, irrespective of development level, were being left behind and that further tools or guidance documentation were required to adapt for the unique challenges faced by these communities [13]. In recognition of these issues, the WHO released a specific step-by-step guide for WSP implementation for small, community-managed water supplies [12] and a field guide to support practical application [14].

Although the recent WHO/International Water Association (IWA) report (2017) indicates that the uptake of WSPs in low- and middle-income countries (LMICs) has increased, generally for all supply sizes, there are still concerns that the uptake in some regions, such as Africa [15], is much slower. The reasons for this are unclear, though researchers have shown that a greater understanding of the benefits associated with WSP implementation is required [16]. Therefore, a better understanding of the factors which make a WSP successful is required.

Small drinking water supplies are often defined by criteria such as the population served, the volume of water supplied, the number of service connections, or the technology type of the supply [17]. There is, however, no globally agreed definition. Small drinking water supplies differ from larger supplies typically based on their operational and managerial arrangements [17].

Literature describing enabling environments and success factors for WSP implementation has started to emerge in recent years, with factors such as financial and human resources, regulations, tools, guidelines, training, and efficient monitoring identified by researchers [18,19]. Although factors may overlap regardless of context, the focus of these studies have been on WSP implementation in utility-managed supplies [20], high-income settings [18], or in the global context [19], with a lack of specific focus on LMICs. This is important to address, since the burden of water-related diseases is greater in LMICs than in high-income countries [21]. It is in these settings that WSPs have the potential to provide the greatest improvements from current baseline conditions [11], helping to achieve Sustainable Development Goal target 6.1 to achieve universal and equitable access to safe and affordable drinking water for all by 2030.

Success factor criteria can be defined by their potential impact, temporal association, the scale of the population reached, and agreement between stakeholders that they are indeed a success factor [22].

The aim of this paper is to identify success factors which assist in the implementation of WSPs in small drinking water supplies in low- and middle-income settings. Based on an extensive literature search, good practices and examples which have led to positive outcomes were identified. In the process, the challenges specific to these settings were also noted. WSP success is dependent on input from a number of stakeholder groups. Therefore, examples and recommendations at both national and

local levels were considered. The findings were analysed to identify trends across various geographic regions and levels of development. The results were compared with existing literature for high-income countries and urban settings to understand the commonalities and differences depending on the implementation settings. This work is important in order to improve and assist with future WSP implementation in small drinking water supplies in LMICs.

## 2. Materials and Methods

A systematic literature search was carried out between January and March 2020. The search criteria included articles available in English published since 2004, when the recommendation of the WSP methodology was included in the WHO Guidelines for drinking water quality [1]. Initial searches were not based on the country development level, as some studies cover both high- and low- or middle-income settings. Income level was applied during the processing of the retrieved articles.

The search terms "impact", "success", "enabl\*", "experience", "legislat\*", "regulat\*", "water safety plan", and "risk management" were used to search for relevant publications in two databases: SCOPUS and Web of Science. The WSP portal [23] and the bibliographies of the papers returned in the search were reviewed for additional relevant published and grey literature.

The papers included in the study were identified based on the Preferred Reporting Items for Systematic Reviews and Meta-Analyses (PRISM-A) methodology [24]. The Development Assistance Committee list of Official Development Assistance (ODA) recipients for 2020 was used to assign low- or middle-income levels [25].

Inductive and deductive coding methods were used [26] using the NVIVO 12 software package to identify key themes. Pre-determined themes (codes) were created based on the researcher's prior understanding of WSPs. These themes included "communication and collaboration", "training and expertise", "compliance and political support", "data systems", "stakeholder engagement", "supporting structures", "community engagement", "record keeping and documentation", and "water quality monitoring". The themes were adapted and added to as the literature was coded.

## 3. Results

The search returned 405 unique publications. Following a review of the titles, abstracts, and subsequently full texts, 48 publications were included in the study (Figure 1). These publications were related to rural or community-managed supplies, including small island communities, in low- or middle-income settings, and were focused solely on drinking water and not bathing, environmental or hospital settings.

Selected publications included 26 case studies or commentary pieces, 11 research articles, 8 guidance documents or supporting resources, and 3 review papers. Publications focused on a variety of technologies commonly used in small drinking water supplies, including hand-dug wells, hand-pump boreholes, household water treatment, water refilling stations and rainwater harvesting.

Search results included case studies from ODA recipient countries classed by the Organisation for Economic Co-operation and Development (OECD) (2019) as "least developed", "lower middle income" and "upper middle-income" and spanned Africa, Asia, Latin America and the Caribbean and Pacific islands. A map showing the geographic spread and density of the publications included in the review is provided in Figure 2. The highest density (30%) of publications by ODA recipient country was around the southern Asian area; countries including Nepal, Bangladesh, India, Sri Lanka and Bhutan. Following this, several publications from the Philippines, Indonesia, the Pacific Islands, South Africa and ODA recipient countries in the regions of Sub-Saharan Africa and South East Asia were included. Publications from ODA recipient countries in the South American and Central Asian regions were sparse, with only 8% and 9% of all the included publications from these regions, respectively. There were no publications identified from ODA recipient countries in Eastern Europe. 47% of the

publications were from countries listed as OECD least developed, with 23% and 22% of publications listed as lower and upper middle-income countries, respectively.

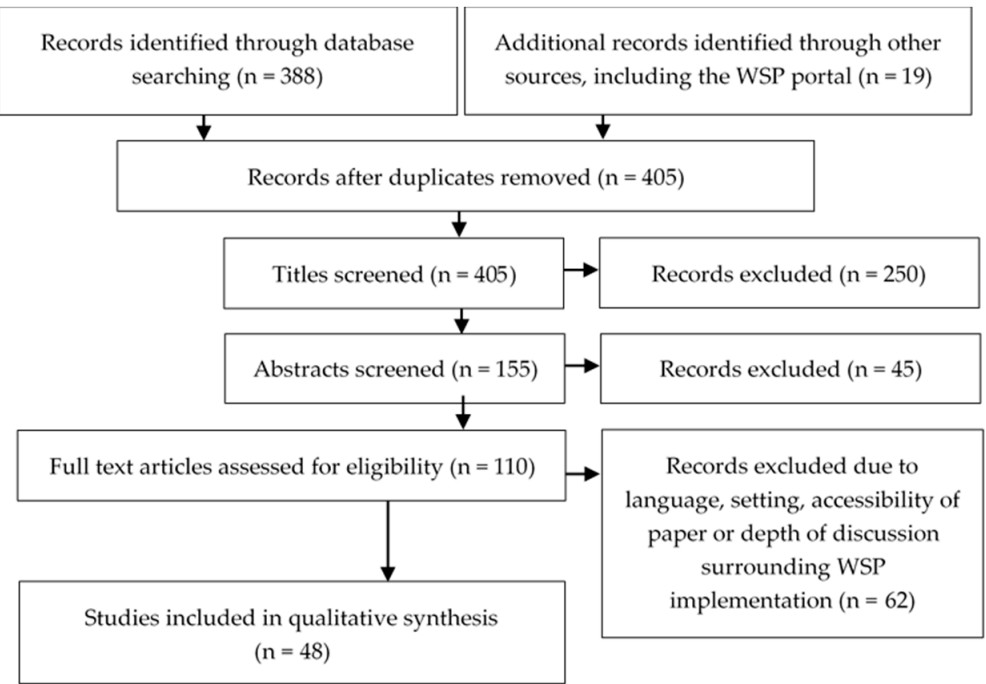

**Figure 1.** PRISM-A search results.

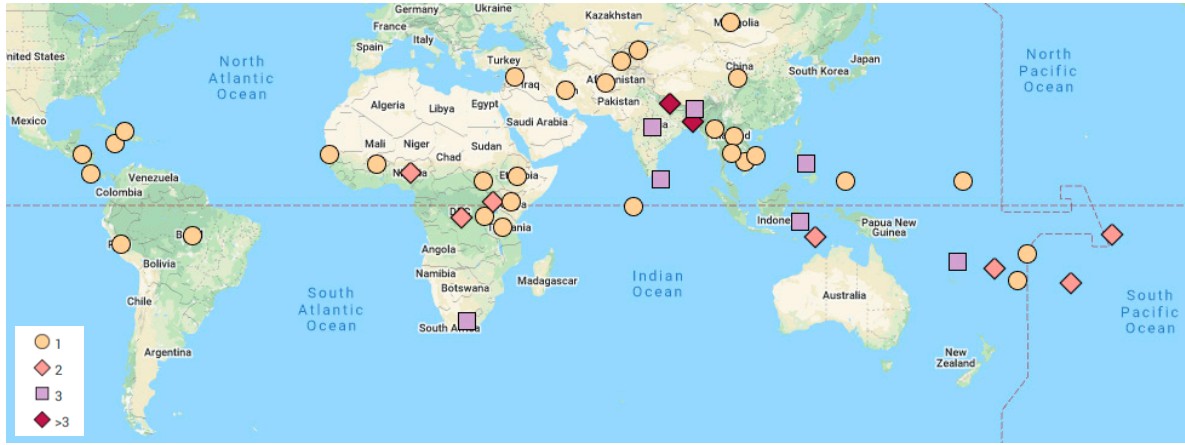

**Figure 2.** A map showing the number of publications included in the review per country (created using GoogleMaps^TM 2020).

*3.1. Defining Success Factors*

The search highlighted a lack of documented literature focused specifically around success factors. Seven publications were identified as stating recommendations or frameworks for effective WSP implementation. These recommendations were collated, and a summary is provided in Table 1.

**Table 1.** Recommendations or frameworks identified from the literature as contributing factors to successful WSPs in LMICs.

| Theme |
| --- |
| Provision of training and developing technical capacity [27,28]. |
| Simplification of tools [27], supporting strategies for community and rural supplies [29], and compulsory supporting programs [30]. |
| Motivating community level uptake via supervision and encouragement from external bodies (preferably government) [27], community-based management of the WSP [29]. |
| Integrating into existing water, sanitation and hygiene (WASH) programs [27], user hygiene collection practices and promotion schemes [31], the sanction of unruly behaviour, and source hygiene management rules [30]. |
| Linking to other ongoing in-country initiatives [30]. |
| Establishing financial and technical assistance for permanency [27], economic support [28]. |
| Baseline water quality data collection to understand WSP outcomes [27]. |
| Institutional support [28]. |
| Social and cultural considerations [28]. |
| Environmental and health benefits [28]. |
| Promotion of water treatment and water quality monitoring at the community or household level [27], monitoring and management by caretakers and water users, and the availability of related training and tools [31]. |
| Champions and stakeholder commitment, including anyone involved in the provision of safe water [29]. |
| Recognising and ensuring realistic timescales for improvements [29]. |
| Developing performance targets, including key performance indicators, documentation, and record keeping; procedures for evaluating the WSP [32]; setting and monitoring targets to build evidence to identify the effectiveness of WSPs [11]. |

Success factors were extracted from the remaining literature based on the language used and suggestions or conclusions made by authors. For example, "stakeholders need to adapt the WSP to their local context" [33], "capacity building ... play a strong role in supporting WSPs" [34], and "there is a need for a systematic approach with multiple stakeholder involvement to develop a comprehensive and effective WSP strategy" [35]. The results of this literature analysis were combined with the findings from Table 1 to create a set of twenty-one success factors. These factors, along with the number of publications in which each factor was identified, are provided in Figure 3.

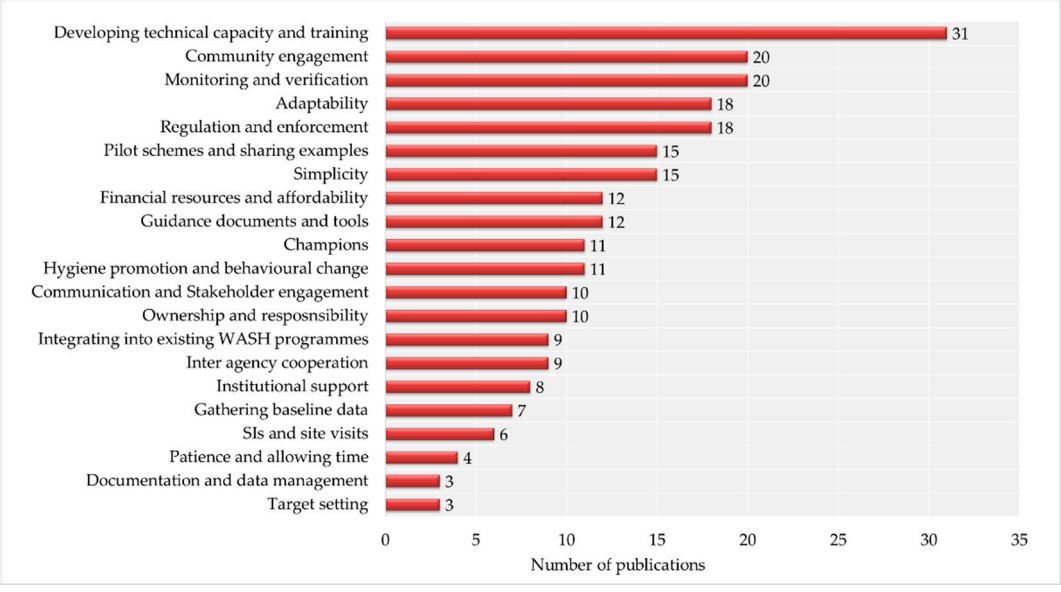

**Figure 3.** Success factors and the associated number of publications citing each factor.

Case studies and examples to support the success factors in Figure 3 were identified during the review. A summary of examples which highlighted that the identified factors were in keeping with the definition of a "success factor" (i.e., their potential impact, temporal association, scale and stakeholder consensus [22]) are provided in Table 2. Examples are grouped by their similarity in the theme of which related success factors are attributed. A number of these examples have overlapping success factors, with "Developing technical capacity", "Community engagement", "Sharing examples", "Affordability" and "Monitoring and verification" noted several times.

**Table 2.** Examples of how factors have contributed to the success of WSPs.

*Developing technical capacity and training*: Expert input from national and international resources [11,36]; development of national training tools and technical resources from sources such as external donors, charitable organisations, and district offices [37,38]; extensive training of sector professionals including caretakers [11,36,38]; training in the use of simple-to-use $H_2S$ tests [32,35,39]; teaching children through games [40]; creation of WSP experts [38]; using trained sanitary inspectors to coach operators to refine and improve WSP quality [41].

*Community engagement*: Community-driven initiatives supported by non-governmental organisations (NGOs) [35] such as community campaigning, school programmes, audio-visual campaigns, public service announcements, leaflets, and posters [42]; trialling non-traditional approaches to engage local communities and stakeholders with support from partners and in-country NGOs [35]; community readiness for change [43]; providing stakeholders with the relevant information to their roles and responsibilities and not overwhelming them with information [40]; empowering often untrained unremunerated communities and workers [35,44].

*Monitoring and verification*: Using qualitative measures as well as quantitative targets; setting simple, low-cost targets such as behavioural changes in sanitary practices and visual checks [32]; using sanitary inspections (SIs) in combination with water quality testing [45].

*Adaptability*: Adapting to the local cultures, contexts, and beliefs—for example, devising separate male and female maintenance and monitoring plans [46] or having a female cashier because, as a woman, she is closer to and is more trusted by the community [42].

*Pilot schemes and sharing examples*: WSP-related networks or platforms [47], international knowledge exchange, training, and technical assistance [36]; pilot trials to identify effective methodologies, viability, and proof of concept [35,41,48]; using successful examples as demonstration sites [37].

*Financial resources and affordability*: Examples of financial support include from NGOs—i.e., WaterAid Nepal [13]; foreign government aid (i.e., Finland foreign affairs to Tajikistan) [49]; grants from the EU and from the in-country ministry of health [50], and from WHO/Australian AID [44].

*Hygiene promotion and behavioural change*: Hygiene awareness and behaviour campaigns for water management committees, caretakers, and communities [11,13,35], including "learn by play" initiatives [40,51].

*Integrating into existing WASH programmes*: Integrating into existing government initiatives [41].

*Inter-agency cooperation*: Institutions, including universities, NGOs, the country's government, and the WHO, working together to implement WSPs [52].

*Institutional support*: Using NGOs to support WSPs—e.g., to roll out, gather feedback, and monitor WSPs [11]. For example, Tearfund now use WSPs as the main method of managing community-based water supply projects and have integrated WSPs into all of their WASH programmes [46].

*Gathering baseline data*: Using pre-existing structures and data sources—i.e., census data [41].

*Documentation and data management*: Data structures which can handle the increased volume of data as monitoring increases [41].

*Target setting*: Developing a national strategy with health-based targets from the outset [35].

### 3.2. Success Factors by Setting

The results from Figure 3 were analysed by geographical region and then by level of economic development to explore trends within each.

Figure 4 shows the percentage of publications stating each of the 21 success factors based on their geographic region. Some variability between the success factors and geographic regions was noted. At least 50% of all publications, regardless of their geographic location, denoted development of technical capacity as a success factor. In all regions, monitoring and verification, regulation and enforcement, adaptability, and communication and stakeholder engagement were identified as success factors. Most success factors were identified across every region, indicating that there are not necessarily huge variations in factors based on geographical location. The only exceptions to this were target

setting, which was not noted in any publications from Sub-Saharan Africa, and documentation and data management, which were not noted in any publications from SE Asian/Pacific Islands and China.

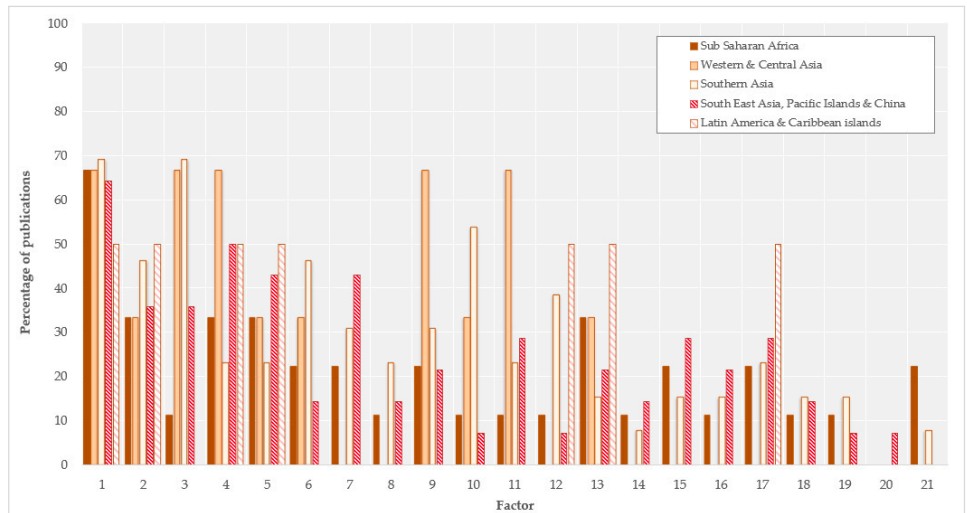

**Figure 4.** Success factors per geographical region (1—developing technical capacity and training; 2—monitoring and verification; 3—community engagement; 4—regulation and enforcement, 5—adaptability; 6—simplicity; 7—pilot schemes and sharing examples; 8—guidance documents and tools; 9—financial resources and affordability; 10—hygiene promotion and behavioural change; 11—champions; 12—ownership and responsibility; 13—communication and stakeholder engagement; 14—inter-agency cooperation; 15—integrating into existing WASH programmes; 16—institutional support; 17—gathering baseline data; 18—SI and site visits; 19—patience and allowing time; 20—target setting; 21—documentation and data management).

Factors which were independent of the level of economic development (Figure 5) included developing technical capacity and monitoring and verification. Publications from the least-developed countries placed a greater emphasis on factors around engaging with the community, including identifying ownership and responsibility, adapting to the setting in question, keeping the WSP simple, and focussing on hygiene promotion and behavioural change. As development level increased, greater emphasis was placed on the supporting structures of regulation and enforcement, financial resources, communication and stakeholder engagement, and gathering baseline data.

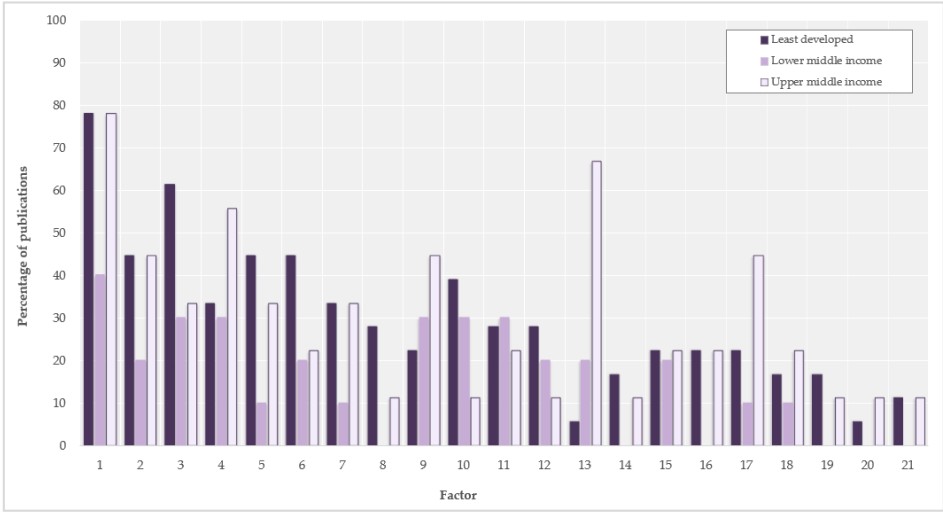

**Figure 5.** Success factors per development level (factor numbers denoted are the same as in Figure 4).

## 4. Discussion

WSP implementation in small drinking water supplies in LMICs has been demonstrated to improve the quality of drinking water and, in turn, public health [11]. This study compiled a set of success factors for WSP implementation in LMICs, both for local and national level implementation, to generate an enabling environment. Certain factors, such as developing technical capacity, monitoring and verification, and regulation and enforcement, are noted throughout the literature, whilst the themes of simplicity and adaptability cut across several of the success factors.

The diversity and overlap of the twenty-one success factors identified during this review, as well as the challenges noted, highlight the complexity of WSP implementation. The results show that, for successful WSP implementation, a combination of success factors at the local and the national level are necessary, and that stakeholder groups at both levels should therefore be involved. As the factors were developed based on the inclusion of documented experiences, the ability to achieve the factors in these settings is considered to be realistic, although certain settings may find some more or less challenging to achieve depending on external influences. As with the ongoing, evolving WSP process, these factors are to be considered as a continual process, not activities to achieve and then move on from.

The following sections provide national and local level recommendations for how implementers, regulators, and other stakeholders can use these factors during WSP implementation in LMICs based on the "success factor" definition stated previously [22].

### 4.1. Strategies for Implementing WSPs

The results indicate that WSPs have been introduced to rural settings through government programmes, collaboration with non-governmental organisations (NGOs) [28,41] and national level conferences [11]. WSP implementation driven by donor requirements and support from professional and industry associations has also been suggested [53], however there is little evidence of these methods being used to drive WSP implementation in rural LMIC supplies. In countries where WSPs have been driven by the government or NGOs, WSPs have been embedded as business as usual indicating that this support has a positive impact on implementation. For instance, following the results of pilot trials in rural Tajikistan, Oxfam GB started to include the WSP process into their projects [2]. The relatively large proportion of case studies identified during the literature search, may indicate that for small water supplies in LMICs, a bottom up approach is often taken, especially where no regulatory requirements for implementation exist.

Running pilot schemes prior to mainstream WSP implementation helps to verify and demonstrate the effectiveness of methodologies, especially in the specific context in question. Furthermore, pilot schemes can help to demonstrate benefits and challenges and to provide lessons learned and capacity for scaling up [13,27,31,35,41,54]. Driven by both governments and NGOs, pilot schemes have been carried out in small drinking water supplies in The Cook Islands, Tonga, Vanuatu and Palau [35], Congo [41], Bangladesh [11] and Nepal [40]. Despite this widespread application of pilot schemes, the challenge of upscaling these to cover all supplies is apparent [11,36,38,41]. Strategies for scaling up, which include policy, training, and national strategies, are required not just for small drinking water supplies but for large supplies in LMICs and high-income countries [48,53].

### 4.2. Developing Technical Capacity and Training

The relevant training of appropriate groups and individuals, from water operators to regulators, is required to develop capacity. Many other success factors identified in this study—for instance, sharing examples, communication and stakeholder collaboration, and hygiene promotion—will support this upskilling of stakeholders. Developing technical capacity has been identified in this study as the most commonly referenced success factor regardless of the development level or geography.

Improving technical capacity at the national level can help to provide focus and resources, including training resources, to assist the WSP process.

As small drinking water supplies are often managed by non-technical, local people, developing and retaining this technical capacity can be hugely challenging. The water sector, in general, suffers from staffing constraints, especially in rural areas [34,55]. Further, in those cases where there is sufficient human resource, WSPs may require a change to the skill profile of the type of work needed. For instance, a shift from a laboratory-based skillset to an inspector-based skillset [56]. Financial limitations can restrict the ability to "buy-in" the required capacity, and the remoteness of rural supplies, physically, technologically and linguistically means that innovative ways to reach these localities are required. Translations of guidance material and supporting tools or resources both into local dialects and into non-technical language could be one such strategy [38,57]. Regardless, communities require top-down support from institutions or their government.

### 4.3. Monitoring and Verification

Monitoring is noted as being "the crux of WSP" [46]. Although small drinking water supplies often have access to limited resources, this should not be used "as an 'excuse' for not implementing WSPs" [56]. The use of field-based tools such as $H_2S$ and tryptophan-like fluorescence have been piloted in a number of countries (e.g., [58]) in anticipation that this may improve the ability of communities to conduct high-level assessments of risks to their supplies. Associated supporting structures around monitoring, such as the availability of laboratory capacity and activities around storing and interpreting data, must be considered carefully, as data collection itself is only one step in the process. Taking action to improve supplies based on monitoring data requires sufficient finances or resources to be available. National level stakeholders can support monitoring programmes by leveraging resources as well as focusing on infrastructures, such as road networks, which can assist with monitoring programmes.

Sanitary inspections (SIs) are proposed as a form of simple WSP. Although not encompassing of the behavioural and management elements that a WSP includes, they are very useful tools to monitor and verify risks to a supply [59]. As they have been widely used for several decades, these are tools which may be familiar to a number of small drinking water supplies and so should not require a great deal of additional work to incorporate into the WSP. The use of SIs at the local level should therefore be encouraged.

### 4.4. Community Engagement

The first step of the WSP methodology for small drinking water supplies is to engage the community [12]. It therefore goes without saying that community engagement and empowering people at the local level are critical aspects of the WSP implementation process. Often, communities see the value in the WSP process and have the drive to improve their supplies, but they just require the means and support to do so [36]. Tools which take into account local cultures, beliefs, and technical awareness will improve engagement with the community and are therefore particularly important in these settings where the community often manage the supplies. On the other hand, engaging with the community will provide a better understanding of the cultures and practices in that locality to make the WSP more fitted to the setting.

### 4.5. Adaptability

All WSPs should be adapted for the individual situation. The complexity of each stage or activity in the WSP process must be appropriate to the stakeholder and technology it relates to. Small drinking water supplies are often less complex than large supplies, and it is especially important in rural supplies that this is reflected in the WSP [32,33,36], as this can make the WSP more "meaningful and accessible for communities to use" [11].

WSPs should be set up so that the community or local group maintaining the WSP can easily amend it based on changing environments and situations. It is therefore crucial that the local WSP

team has the capability and the technical know-how to make appropriate modifications to keep the WSP relevant and up to date. As experience and knowledge of the supply and WSP process is gained, the WSP methodology can and should evolve to better suit the location [40].

### 4.6. Sharing of Information

The use of pilot schemes and sharing examples were identified in this study as methods to assist with implementation and wider scale roll-out. The broader notion of sharing information, internationally, nationally, and locally, is incredibly beneficial, particularly when it comes to developing technical capacity.

Cascading training from large, national level, disease control centres to operational utilities, and larger utilities mentoring smaller ones are effective methods of building capacity in utility-based settings [36,60,61]. Identifying and collaborating with similar in-country institutions relevant to small drinking water supplies in this way could help to build technical capacity, although consideration is required for the financing and complexity of technology used. Approaches involving the training of trainer's model are effective ways of assisting with developing technical capacity when scaling up from pilot projects and in helping to reach a larger number of people [33,34].

### 4.7. Affordability and Financial Support

The UN-Water Global Analysis and Assessment of Sanitation and Drinking Water 2019 Report report highlighted that the lack of human and financial resources are one of the key factors constraining the implementation of national WASH policies and plans, and that there are huge funding gaps between what is needed to meet WASH targets and available funds [62]. Studies which seek to understand and evaluate implementation of WSPs with regards to financial costs are limited [20]. Initial studies have shown that just following the WSP process can lead to a more cost-effective use of limited resources [6], with a cost benefit ratio of 1:6 in some cases [35].

Establishing sustainable financial assistance from the outset, especially for community-managed supplies which may already have limited financial budgets, is critical for successful WSP implementation [27,28,37,63,64]. WSPs should not put an increased strain on communities who may already struggle financially. Often, any WSP-related costs will be transferred to the community, so the use of existing processes or cost-effective methods can help implementation and sustainability [11,28]. External or national level funding sources may be required to assist implementation (see Table 2 for examples).

Although the emphasis should be more on following the WSP process than the funding [37], regardless of whether financial barriers are true or perceived, the issue must be raised and tackled. Even perceived financial blockers may inhibit WSP implementation in areas of elevated poverty, and users, especially in communities, must therefore be given various financial choices [30].

### 4.8. Guidance Documentation or Tools

There is an abundance of guidance documentation or tools available from the WHO and IWA and others regarding the implementation of WSPs in both large supplies and small drinking water supplies [4,12]. These documents have been used as the basis for WSP implementation in many small water supply settings, including in the Pacific Islands [35], Uganda [65], Congo [41], Nigeria [30] and the Philippines [66]. They have also been adapted for the local context—for example, community checklists in Nepal [13], SIs in the Marshall Islands [5], and the national level Department of Health guidance in the Philippines [54]. Consideration for easy to understand and practical guidance documentation in the local language or dialect is required [2].

### 4.9. Hygiene Promotion and Behavioural Change

Hygiene education and promotion campaigns around hygiene and sanitation practices should be integrated into the WSP process. Without prior WASH education, the uptake of WSPs has been

noted to be limited, whereas inclusion has been identified as improving WSP practices by 20% [67]. Attention should be given to resource protection and sanitation practices to minimise the risk to source water quality, the first stage in the water treatment process. The WSP approach needs to be looked at holistically, as part of any WASH programme, and not in isolation.

## 4.10. Responsibility and Ownership

Although ultimately it is the local community who must acknowledge and accept responsibility for the WSP to be successful and sustainable [13,39], stakeholders, such as government ministries and NGOs, are required to facilitate and support implementation. The management of rural supplies, in general, is a problem and one which requires addressing [36].

NGOs often have a great deal of experience in terms of implementing and running projects similar to WSPs [46] and are particularly important stakeholders in the implementation of WSPs in rural settings [11,34,35,41,42,68].

## 4.11. Integration into Already Existing Structures

WSPs can be introduced as stand-alone activities or, better still, integrated into already-existing WASH programmes [46]. Embedding or making use of existing resources, platforms, datasets, and committees, including government ministry bodies [63], can be a beneficial way to reduce the volume of additional work, or duplication of work, and the overall resources required. Building on what has already been achieved and integrating into existing programs or activities has been shown to be cost-effective and sustainable [11,29] and can help to streamline activities. For instance, the team may have already been assembled and training activities may already be being carried out [46]. In some cases, monitoring and verification strategies or controls may already exist, in which case just a review is required to ensure suitability as part of the WSP [38]. It is important not to overcomplicate, duplicate, or make changes to a setup which may already work, but to embed these into the WSP especially in resource-limited locations. Caution needs to be taken when using pre-existing structures, however, as this may exacerbate already existing prejudices and structures which make equity issues worse [69]. Additional work to identify how WSPs can fit in with pre-existing WASH measures is required [38].

## 4.12. Institutional Support

Careful consideration for which organisations may be able to support implementation and ongoing WSP-related work can assist with WSP implementation. NGOs, in particular, have contributed to the collection of a wide range of available guidance material. For instance, Tearfund and WaterAid have produced guidance documentation aimed at disaster management teams and community-level management for water security [46,70]. WSPs should incorporate any relevant existing work or activities carried out by NGOs and should not be a repeat of some of the large volumes of work they already do [13]. NGOs can help to ensure equity in WSP development—for example, the consideration of gender and vulnerable groups [35]—and can also provide financial support for WSP implementation [13].

## 4.13. Variation of Success Factor by Setting

The results indicated some variability in success factor with geographical location. The most common factors, such as developing technical capacity and community engagement, were identified in all regions, however the less frequently referenced factors, such as target setting and documentation, were only identified in the Sub-Saharan African, Southern Asian, South East Asia, Pacific Islands, and China regions. Success factors are not necessarily different based on geography, but these findings highlight the need for further published research in the less well-studied regions of Western and Central Asia, Latin America, and the Caribbean Islands.

### 4.14. Comparison of Studies with Those in Urban and High-Income Settings

Success factors in urban and high-income contexts are also understudied, although studies into the creation of enabling environments, which also plays a big part in supporting WSP implementation, are starting to emerge [20,71]. An enabling environment has been defined as "a favourable culture of internal coordination and communication; policy and institutional behaviour that guides behaviour of water service providers with clear and enforceable service standards and resources to provide effective water services" [72].

Some factors identified in this study have also been noted as enabling environments in high- and low-income urban settings, indicating some continuity and cross-over of beneficial activities regardless of the setting/income level. These include the use of guideline documentation, tools, national training materials, training facilitators, international collaboration, and the role of pilot studies and sharing examples [18,71,73]. Knowledge sharing, stakeholder involvement, training, champions, and investment in monitoring and ongoing WSP verification are also noted as WSP enablers in low-income urban settings [15]. Although there may be crossover in the factors, the individual application will be different for high-income and urban settings to small drinking water supplies in LMICs. For example, the small LMIC supplies will involve greater NGO involvement, whereas urban and high-income settings will have greater engagement with other types of stakeholders.

Sharing the characteristics of remoteness and limited human and financial resources, and therefore restricted outreach of support, small drinking water supplies, regardless of a country's income level, are "disproportionately problematic" [37,73]. Although some factors have been noted as similar regardless of context, small drinking water supplies in LMICs are often faced with additional challenges when trying to facilitate these enablers. For example, the scarcity of documented experiences, specifically in small drinking water supplies in developing countries [11], means that sharing examples could be challenging. Issues around the availability of baseline data are more common in lower-capacity settings [74], and therefore more resources and focus are required in this area of work [27].

Identifying the financial support required for WSP implementation, both short- and long-term, and the associated sources of funding is also identified as necessary in high-income settings, especially for small drinking water supplies, although the funding structures may differ between the contexts [18]. Small drinking water supplies generally have lower per capita operation and maintenance costs than larger supplies; however, the financing of small water supplies in both high-income countries and LMICs is an area which requires greater focus [75].

### 4.15. Regulation and Enforcement

There are conflicting views surrounding the role of regulation as an enabling environment or success factor. This has been identified in this study but also by others in high-income settings [71]. Supportive legislation can facilitate and form a key component of WSP implementation, particularly in small drinking water supplies, for instance by helping to build capacity and mobilise resources [18,34,35,64]. The role of supportive regulation as opposed to strict regulation is increasingly being discussed and agreed as the appropriate way to regulate rural supplies in general [76]. Some publications note that promotion of WSPs by governments, including new standards and regulations, will increase the uptake of implementation [36], although with a caveat that they should not become "just another compliance issue" [64]. Other publications note that although a sound regulatory framework should be perceived as an enabler, their absence should not be perceived to be a blocker or to delay WSP implementation, especially as legislation can take years to implement, whereas operational procedures can be developed on shorter timescales [63,77]. "Self-help is the best help . . . 90% of the problems can be solved at community levels. Government agencies are just the facilitators and communities are the implementers" [44].

Decentralisation of governments is becoming a more common way of regulating rural supplies [78]. This supports and further enables communities to take ownership of and responsibility for their

individual supplies. Enforcement of legislation can be a challenge, especially in low-income settings where governments have other priorities [56,77].

The majority of the articles included in this study were published by researchers and public health bodies, with only a few publications by government ministries and departments. This could skew the perception of the work carried out or struggles faced by regulators. Further work into the role of regulation in WSP implementation, regardless of setting, is required.

### 4.16. Areas for Further Work

The skew of included publications towards case studies highlights the limited volume of original research on the topic of WSP success factors in LMICs. Although this bias has enabled the success factors in this study to be developed based on a reflection of experiences and has enabled the inclusion of geographies which are not represented by published work, the lack of original research publications indicates further work and that studies into this topic are required.

Due to the importance of community engagement, a community's readiness for change must be addressed. There is limited focus on this aspect of WSP implementation in the literature [43]; hence, further work is required to identify strategies for preparing communities for potential change.

The use of WSPs, especially for small drinking water supplies in LMICs, in the context of emergency planning is poorly documented [36]. The effects of climate change are acknowledged as one of the most significant emerging issues for the WASH industry [38], and the impacts disproportionately affect low- and middle-income communities [79]. Although recent studies demonstrate how climate change can affect different community supplies [80,81], guidance on and examples of how to incorporate these findings in to WSPs, specifically in small drinking water supplies in LMICs, are limited [81].

A finding of this study is the lack of published studies, with a focus on success factors and enabling environments for WSP implementation, both generally but also especially for small drinking water supplies in LMICs. A recommendation from this work would be a dedicated data collection to assist the understanding of what factors make a successful WSP and best practices for WSP implementation.

This study has identified less published literature for case studies in the South American, Eastern European, and Central Asian regions (Figure 2). Although this may be due to language limitations or simply a lack of publications, not necessarily reflecting the actual work being carried out, future researchers may wish to consider this in subsequent studies.

### 4.17. Limitations of the Study

It is recognised that not all countries who have reported the use of WSPs [10] were included in this review, and therefore this study only encapsulates a snapshot of WSPs which have been documented by published literature. For instance, the study noted a lack of publications for OECD countries in the Eastern European and Central Asian regions, although regional documentation indicates that a great deal of work is being carried out in countries such as the Republic of Moldova, Georgia, and Tajikistan [2].

Refining searches to the English language could have impacted the diversity of the literature found. In addition to the many languages and local dialects spoken across LMICs, Portuguese is commonly spoken in parts of Africa and South America, Spanish in South America, French in parts of Africa, Arabic in parts of Africa and Central Asia, and Russian in the Central Asian regions. Although English-only reviews do not necessarily lead to systematic bias in results [82], it seems pertinent to conduct future studies in other languages, especially as one of the findings was that documentation should be translated into local languages and dialects. To some degree, these variations in languages correlate with the lack of inclusion in this study from these locations (Figure 2). The authors have tried to reduce the impact of this limitation with the inclusion of a wide variety of case studies.

## 5. Conclusions

This study has produced a set of twenty-one success factors for WSP implementation in small drinking water supplies in LMICs. In the process of doing so, the challenges, including financing and technical support, which must be acknowledged are noted. The notion of a supportive environment is emphasised throughout. It is anticipated that this information may assist the many stakeholders, including implementers and regulators, whose involvement is required for success, with areas to focus on to improve the spread and sustainability of WSPs.

The degree of influence each factor has on WSP implementation indicates that the most important success factors in these settings are developing technical capacity, community engagement, monitoring and verification, hygiene promotion, and simplicity. The sharing of knowledge and examples and the use of relevant or tailored guidance documentation are hugely beneficial activities to help build technical capacity as well as support and momentum for implementation. Work which is already being carried out should be built upon, with a WSP incorporated into such activities so as not to duplicate efforts, waste limited resources, or overwhelm communities. Organisations which can help facilitate the roll out and long-term sustainability of WSPs must be identified as early as possible, with NGOs recognised as one of these key players.

A comparison of the findings from this study with findings from urban and high-income settings indicate that some of the factors identified, such as adaptability, communication, and stakeholder engagement, are applicable regardless of setting. Nonetheless, a consideration of how these factors may be affected by the individual context is required to ensure appropriate application, as their application is different based on the context—i.e., NGO support is not as relevant for small drinking water supplies in high-income countries. The degree of overlap across these key themes reflects the complexity of challenges faced by communities in managing their water supplies. This study highlights that the issues of financing and regulating, as well as the overall management of small drinking water supplies, are not unique to WSPs.

The findings of this study confirm that, although there are positive indications of the widespread global uptake of WSPs, the success factors and enabling environments, especially in LMICs, are understudied. Although the benefits of WSPs are noted, whether these are long-term, sustainable changes remains to be seen.

**Author Contributions:** Conceptualisation, J.H. and K.P.; methodology, J.H.; writing—original draft preparation, J.H.; writing—review and editing, J.H., K.P., B.R., D.J.L., T.M., K.O., R.K. and S.J.H.; funding acquisition, B.R. All authors have read and agreed to the published version of the manuscript.

**Funding:** The APC was funded by The German Environment Agency (UBA).

**Conflicts of Interest:** The authors declare no conflict of interest.

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
