# Peer review of "Success Factors for Water Safety Plan Implementation in Small Drinking Water Supplies in Low- and Middle-Income Countries"

_resources, doi:10.3390/resources9110126_

Round 1
Reviewer 1 Report
The paper makes an exhaustive analysis and is well supported by the extensive bibliography consulted.It is considered that it can be published after minor revisions of detail (for example, Figure 2 is not mentioned in the text, although there is a reference not found in line 144).
The only negative aspect is the use essentially of English-language bibliography. Note that in much of Africa, for example, Portuguese or French are spoken. Authors should make an effort to overcome this limitation in future publications.
Author Response
Comment 1: Figure 2 is not mentioned in the text, reference found in line 144
Response 1: Additional wording has been added in Line 146 saying ‘A map showing the geographic spread and density of publications included in the review is provided in Figure 2’.
Comment 2: The only negative aspect is the use essentially of English-language bibliography. Note that in much of Africa, for example, Portuguese or French are spoken. Authors should make an effort to overcome this limitation in future publications.
Response 2: Thanks for this comment. The authors also recognised this as a limitation and had noted as such in section 4.17. Further wording has been added to this section for further clarity (lines 520 to lines 529).
Reviewer 2 Report
Thank you for asking to review this literature review. It is hard to get excited about the manuscript as it is almost like a manual or policy than a scientific paper. There is no question that the data included are of value without major new scientific additional knowledge to the field.
- The method for searching and including the papers are very good.
- However, the chosen papers are heavily skewed towards care studies. I believe this is a major limitation. I am not sure why the authors did not choose to select the study design in addition to the key words
- figure 2 is not useful in my opinion. It is self report of success by the authors of selected papers rather than any analyzed data by the authors.
- The same criticism apply to Figure 3. However, Figure 3 is probably more informative than Figure 2 as it compiles the references citing a particular factor
- Table 2 could be shorter, clearer and more informative.
- I am not sure if the authors would consider simplifying the language and explaining more the major difference (or additional benefit) in Figure 4
- Discussion: I suggest to remove the 2nd line "the first study to .." It is not an original research and the authors did not review other books or policies that typically compile these interventions.
- The discussion is used to summarize the study rather than discussing controversial issues or relevant results.
Author Response
Comment 1: Lines 36 and 125 typos (focussed should be focused)
Response 1: Done
Comment 2: The method for searching and including the papers are very good. However, the chosen papers are heavily skewed towards care studies. I believe this is a major limitation. I am not sure why the authors did not choose to select the study design in addition to the key words.
Response 2: The skew towards case study papers represent the findings of the search but reiterates the need for further original research into this topic. Wording has been added to line 35 (abstract) and 490-494 (areas for further work) to reiterate this.
Review papers have been included in the study however this skew towards case studies highlights that this manuscript is the first synthesis of studies focused on this specific area (rural supplies in low- and middle-income) and the lack of original research in this field of work. The authors did not want to limit search results by selecting a study design and felt that without the inclusion of so many case studies a true reflection of experiences from broad geographies which would be a greater limitation (see comment 2 from reviewer 1).
Comment 3: Figure 2 is not useful in my opinion. It is self report of success by the authors of selected papers rather than any analyzed data by the authors.
Response 3: The figure is a visual display of the geographic spread of the locations used to develop the success factors in the study highlighting that examples were included from all regions but that the density of information skewed towards some countries or regions. Some analysis of this bias has been included in lines 152 – 155 and the figure has been referenced on lines 509 and 528 of the discussion.
The authors feel this information needs to be displayed and this figure is the most efficient way of doing so.
Comment 4: The same criticism apply to Figure 3. However, Figure 3 is probably more informative than Figure 2 as it compiles the references citing a particular factor
Response 4: This is one of the main results of the paper listing the factors identified. The authors feel this is the most efficient way of displaying the information. The other reviewers did not raise any concerns with the figure and therefore on balance we believe it is worth keeping as it is.
Comment 5: Table 2 could be shorter, clearer and more informative.
Response 5: Thank you for the comment. On reflection we think this is a very valid point. The table has been amended to make it more informative.
Comment 6: I am not sure if the authors would consider simplifying the language and explaining more the major difference (or additional benefit) in Figure 4
Response 6: Further text has been added to lines 189 – 198 for clarity.
Comment 7: Discussion: I suggest to remove the 2nd line "the first study to .." It is not an original research and the authors did not review other books or policies that typically compile these interventions.
Response 7: Removed as suggested.
Comment 8: The discussion is used to summarize the study rather than discussing controversial issues or relevant results.
Response 8: Many thanks for this comment. The authors agree that the discussion could be more critical. Amendments have been made on the following lines.
- 240 – 278
- 293 – 297
- 319 – 321
- 328 – 337
- 353 – 371
- 383 – 387
- 391 – 393
- 418 – 422
- 446 – 452
- 460 – 464
- 469 – 471
- 483 – 486
- 490 – 494
- 499 – 503
- 520 – 529
In addition, lines 533, 546, 550 and 556 of the conclusions have been amended to incorporate changes to the discussion section.
Reviewer 3 Report
Thank you for this very informative paper dealing with an issue that is hightly important for further improvement of water supply in low- and middle-income countries. I suggest no technical corrections, just minor spelling and grammar check.
Author Response
Comment 1: Minor spell and grammar check
Response 1: Lines 36 and 125 typos corrected (focussed should be focused). A further spell and grammar check was carried out, minor amendments have been highlighted in red.
Round 2
Reviewer 2 Report
I appreciate the invite to review the paper. I also went throught the changes by the authors and I approve it as is